# Content-Adaptive and Attention-Based Network for Hand Gesture Recognition

Zongjing Cao, Yan Li and Byeong-Seok Shin *

Department of Electrical and Computer Engineering, Inha University, Incheon 22212, Korea;
zjcao@inha.edu (Z.C.); leeyeon@inha.ac.kr (Y.L.)
* Correspondence: bsshin@inha.ac.kr; Tel.: +82-32-860-7452

**Abstract:** For hand gesture recognition, recurrent neural networks and 3D convolutional neural networks are the most commonly used methods for learning the spatial–temporal features of gestures. The calculation of the hidden state of the recurrent neural network at a specific time is determined by both input at the current time and the output of the hidden state at the previous time, therefore limiting its parallel computation. The large number of weight parameters that need to be optimized leads to high computational costs associated with 3D convolution-based methods. We introduced a transformer-based network for hand gesture recognition, which is a completely self-attentional architecture without any convolution or recurrent layers. The framework classifies hand gestures by focusing on the sequence information of the whole gesture video. In addition, we introduced an adaptive sampling strategy based on the video content to reduce the input of gesture-free frames to the model, thus reducing computational consumption. The proposed network achieved 83.2% and 93.8% recognition accuracy on two publicly available benchmark datasets, NVGesture and EgoGesture datasets, respectively. The results of extensive comparison experiments show that our proposed approach outperforms the existing state-of-the-art gesture recognition systems.

**Keywords:** content-adaptive; attention mechanism; gesture recognition; hand detection





## 1. Introduction

Hand gestures are a form of non-vocal communication that communicate particular messages via visible hand movements. Compared to other physical input devices, hand gestures provide a more natural and convenient way for humans to interact with devices. Computer vision-based gesture recognition is a form of technology that uses a computing module to read and interpret hand movements as commands [1–6]. Computer vision-based gesture recognition technology can be used in several industries, such as interactive entertainment, smart home, VR/AR, and sign language machine translation [7–13]. Due to its wide application in several industries, hand gesture recognition based on computer vision technology has received great attention from researchers in the computer vision community in recent years.

Gestures in a video consist of spatial information in each frame and temporal information in neighboring frames. Capturing motion information between neighboring frames from sequence data is a key challenge for gesture recognition tasks. For example, the operation of zooming in and out of the screen with fingers has similar features in spatial domains, but the temporal information is reversed. Simonyan et al. proposed a method using a two-stream network [14], which combines a color image frame and multiple stacked optical flow frames to predict action classes. As an extension to the two-stream method, Wang et al. presented a temporal segment network (TSN) [15] based on long-range temporal structure modeling. The main idea of TSN is that we can divide the long video into several equal segments, process each segment individually, and then obtain a segment consensus from each segment and perform the final prediction. Despite the performance gains obtained by TSN, this still relies on optical flow features that need to be precomputed. Some

work [16–18] has attempted to use a 3D convolution approach to learn spatial–temporal features in place of optical flow. In these methods, continuous frames from a video clip are stacked and then fed into a network for spatial–temporal modeling. Three-dimensional convolution neural networks (CNNs) have sufficient capacity to encode spatial–temporal information in densely sampled frames, but the huge number of parameters to be optimized leads to expensive computational consumption. Recurrent neural networks (RNNs), which consist of recurrent units, are another kind of neural network used to process sequential data [19,20]. For any moment in the sequence, there is a recurrent unit corresponding to it, which will fuse the input of the current moment and output of the recurrent unit of the previous moment to calculate the output of the current moment. Each recurrent unit of an RNN has forward dependency, i.e., the output of the network at the current moment is jointly determined by the previous sequences. This property allows the past information of the sequence to be continuously transmitted, but also causes a decrease in the efficiency of model operation. Assuming that the length of the input sequence is $n$, the minimum sequential operation required by the recurrent neural network layer is $O(n)$. Since there are $O(n)$ sequential operations, RNNs cannot be operated in parallel, and this sequential modeling approach leads to information loss during long-distance passing.

Since 2017, a new model architecture called transformer [21] has received significant attention due to its performance beyond RNNs in many natural language processing (NLP) tasks. Transformers use a mechanism called self-attention to build sequence models that do not rely on any convolution or recurrent layers. Table 1 compares the minimum number of sequential operations, maximum path length, and complexity for CNNs, RNNs, and transformer layers [21]. As shown in Table 1, self-attention has both an advantage of shortest maximum path length and minimum sequential operation. Therefore, it is attractive to solve machine learning tasks using deep networks based on a self-attention mechanism. Recently, some research [22,23] has attempted to use transformers in the field of computer vision, which traditionally relies on CNNs. Dosovitskiy et al. proposed a vision transformer (ViT) network [24] for image recognition tasks. This is a pure transformer-based network that outperformed the convolutional-based approach. Inspired by ViT and the fact that self-attention enjoys parallel processing, we introduced a transformer-based network for a hand gesture recognition task. This is an entirely self-attentional architecture that predicts gestures by focusing on the whole video sequence information. More specifically, we use a transformer encoder-derived architecture for hand gesture recognition tasks, resulting in a high-accuracy and low-latency model for real-time applications. The architecture of our approach consists of three submodules: a content-based adaptive sampler, a temporal attention-based encoder, and a classification multi-layer perceptron (MLP) header. After obtaining the sequence of tokens mapped from the video data, each token is given a position embedding information gained through learning or fixed absolute encoding. In the same manner as with ViT, we prefix the sequence of the features with a special classification token (CLS). We use the final state of the features related to this CLS token as the final representation of the input video after propagating the sequence through the transformer encoder. Finally, an MLP head processes the CLS token to obtain the final class prediction. Furthermore, we present a dynamic sampling strategy based on content adaptation to efficiently sample the video sequence. We use a sliding window to perform gesture detection on the input video content. When the sampler detects a gesture, it starts sampling evenly according to the time strategy until it reaches the set maximum number of frames. Another role of this sampler is to act as a switch for the encoder. When a gesture is detected in the video sequence, the encoder is activated and starts receiving the input from the sampler. Since most of the time, no gesture appears in the video sequence, there is no need for the feature extractor and encoder to process these frames, which would significantly increase the consumption of the system.

**Table 1.** Comparison of the time complexity of each layer of RNNs, CNNs, and transformer. $n$ indicates the length of the sequence, $d$ indicates the size of the hidden layer, and $k$ indicates the kernel size of the convolution.

| Model | Layer Type | Minimum Sequential Operations | Maximum Path Length | Complexity Per Layer |
|---|---|---|---|---|
| CNNs | Convolutional | $O(1)$ | $O(\log_k(n))$ | $O(k \cdot n \cdot d^2)$ |
| RNNs | Recurrent | $O(n)$ | $O(n)$ | $O(n \cdot d^2)$ |
| Transformer | Self-Attention | $O(1)$ | $O(1)$ | $O(n^2 \cdot d)$ |

The following are the main contributions of our work:

(1) We suggest a transformer-based network for hand gesture recognition tasks, which is a fully self-attentional architecture. The framework classifies gestures by focusing on the entire video sequence without any convolution or recurrent layers. It effectively manages the high number of spatial–temporal tokens that may appear in the video.

(2) We propose an adaptive sampling strategy based on video content to reduce the sampling of gesture-free frames and improve the performance of the model. We use a sliding window to continuously detect the input frame sequence. When a gesture is detected, continuous sampling from the current frame is conducted, with no additional gesture detection.

The rest of this paper will consist of the following. The related studies and background techniques in gesture recognition are discussed in Section 2. Section 3 describes our proposed approach in detail. The experiments and experimental results are shown in Section 4. Section 5 contains the conclusion and suggestions for further studies.

## 2. Related Work

### 2.1. Convolution-Based Methods

Action is composed of two components: appearance information carried by a single frame and motion information between neighboring frames. Capturing motion information between neighboring frames from sequence data is a key step in gesture recognition systems. In reference [14], Simonyan et al. proposed a two-stream approach that combined predictions from a single-color image frame and a stack of externally computed multiple optical flow frames to predict action classes. Two-stream networks are only modeled in a short time and cannot effectively model the temporal structure over a long range. To address this problem, Wang et al. presented a TSN [15] that combined a sparse temporal sampling strategy with video-level supervision to efficiently learn the long-range temporal structure of the video. The TSN's main idea is that the entire video can be evenly divided into several segments, each of which is processed separately. The video-level prediction will then be formed from a consensus among the segments. Subsequently, many studies [16,17,25,26] have employed this idea and designed different deep models for action recognition. However, we observed that this algorithm does not work well for scene-independent action recognition, such as hand gestures. In addition, although optical flow can describe the motion effectively, computational consumption is expensive.

Additionally, 3D convolution and recurrent convolution are two effective methods for modeling temporal structure. In [16], Tran et al. proposed an approach for action classification using 3D CNNs for spatial–temporal feature learning. Compared to 2D CNNs, 3D CNNs are excellent for learning spatial–temporal features. However, the exponential increase in the number of parameters to be optimized for 3D CNNs-based models leads to high computational consumption and the need for more data to train the model. RNNs are neural networks composed of recurrent units [19,20]. The input of the current moment recurrent unit consists of the output of the previous moment and the input of the current moment, which can also be understood as the output obtained from the network's current moment calculation being jointly determined by the previous sequence. For long

videos, long distances can cause information to be lost during delivery, and this sequential modeling approach also limits the parallel computing of the system. To address all these issues, we proposed a transformer-based network for hand gesture recognition. It is a fully self-attentional architecture without any convolutional and recurrent convolutional layers. In contrast to using the segmental consensus-based approach [25,26], our proposed method classifies actions by focusing on the temporal structure of the entire video sequence.

### 2.2. Attention-Based Methods

Since 2017, the transformer has received substantial attention due to its performance beyond RNNs in many NLP tasks. The transformer was first introduced for solving machine translation tasks [21]. Since then, transformer-based sequence modeling approaches have gradually become the state-of-the-art solution for other NLP tasks [27,28]. Traditional self-attention-based models usually rely on RNNs for input representations [29,30]. In contrast, the transformer model does not have any recurrent or convolutional layers and is a model entirely based on attention mechanisms [21,31]. The transformer consists of two submodules: the encoder and the decoder. Both encoder and decoder are stacked with self-attention-based modules. The embedding representations of the input and target sequences are fed into the encoder and decoder, respectively, after adding positional encoding. The encoder of the transformer is composed of $N$ identical layers stacked on top of each other (in reference [21], $N$ is set to 6). Each layer consists of two sublayers: a self-attention sublayer and a feed-forward network sublayer. Similar to the design of the ResNet module, both sublayers are connected using a residual connection layer, followed by layer normalization. Specifically, when computing the encoder's self-attention, the queries, keys and values are taken from the output of the previous encoder layer. Like the encoder, the decoder of the transformer is stacked with $N$ identical layers, with each sublayer also using residual connections and layer normalizations. Unlike the encoder, the decoder adds a sublayer called the encoder–decoder attention layer between the two sublayers of the self-attention layer and the feed-forward network. The encoder–decoder attention sublayer is still based on the self-attention mechanism, so it has the same structure as the self-attention sublayer. As with the input of the encoder self-attention, the queries, keys, and values in the self-attention decoder are also taken from the output of the previous decoder layer. In the new encoder–decoder attention sublayer, the input keys and values are taken from the output of the transformer encoder layer, while the queries are from the output of the previous decoder layer.

Although the original transformer was proposed for solving sequence-to-sequence tasks, it is now widely used in other deep learning areas, such as speech processing, reinforcement learning, and computer vision [31–34]. Inspired by the advanced performance achieved by transformers in the field of NLP, Dosovitskiy et al. [24] proposed a vision transformer network (ViT) for image classification. For an input image, ViT first splits it into $N$ fixed-size patches and then uses the linear embeddings sequence of these patches as the input to the transformer encoder. In ViT, image patches are treated in the same manner as words (tokens) are processed in an NLP application. To our knowledge, ViT was the first approach to apply transformer to computer vision tasks. After this, transformers started to be employed in a wide range of other computer vision tasks. For example, DETR [35] for object detection and VisTR [36] for video instance segmentation have yielded competitive results using transformer-based approaches. Since the transformer has a component based on the self-attention mechanism, it enables the network to capture contextual information from the whole sequence data, which leads to the success of the transformer. However, the computation of self-attention grows quadratically with the length of the sequence data, resulting in the standard transformer-based model being unable to effectively process long sequence data. To address this challenge, Beltagy et al. proposed an improved transformer architecture, called longformer [37]. The longformer has an improved self-attention mechanism, and the operations of this attention mechanism are scaled linearly with the sequence length, enabling it to handle longer sequence data. Longformer's self-attention mechanism

has a time complexity of $O(n)$. Considering its ability to process long sequences and its temporal complexity, we use longformer as our self-attention module.

## 3. Proposed Method

Inspired by the application of the transformer in image classification tasks [24], we introduced a transformer-based architecture for hand gesture recognition tasks. The proposed network uses only a self-attention mechanism and standard feed-forward neural networks without any recurrent units or convolutional operations. The architecture of our proposed network is illustrated in Figure 1. The network is composed of three submodules: a content-based adaptive sampler, a temporal attention-based encoder, and a classification MLP head.

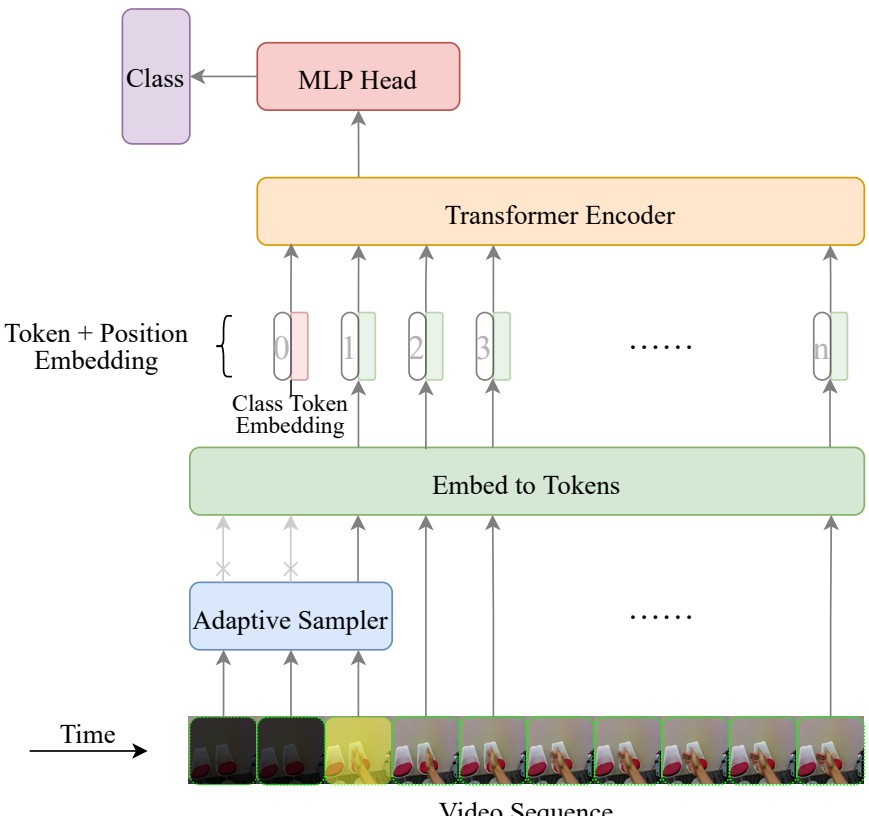

**Figure 1.** The architecture of the gesture recognition network based on content adaptation and attention.

### 3.1. Content-Adaptive Sampler

Gestures can be roughly categorized into two components: appearances and motion. Understanding the motion in gesture videos relies heavily on long-range temporal information. Learning video representations that capture long-range temporal information is a pivotal challenge for gesture recognition. To use the dynamic information from the whole video for video-level prediction, a sparse temporal sampling strategy is currently the dominant approach. The main idea is to divide a video into k segments evenly and then randomly select n frames from each segment. We visualized the sampling results of the sparse temporal sampling strategy, as shown in Figure 2. Figure 2a presents the sampling results in a NVIDIA dynamic hand gestures (NVGesture) dataset when $k = 16$ and $n = 1$. Figure 2b indicates the results when $k = 32$ and $n = 1$. We found that when $k$ is set too low, the sampling results do not effectively represent the whole video; when $k$ is set too high, too many frames without gestures are sampled, increasing the input length of the model.

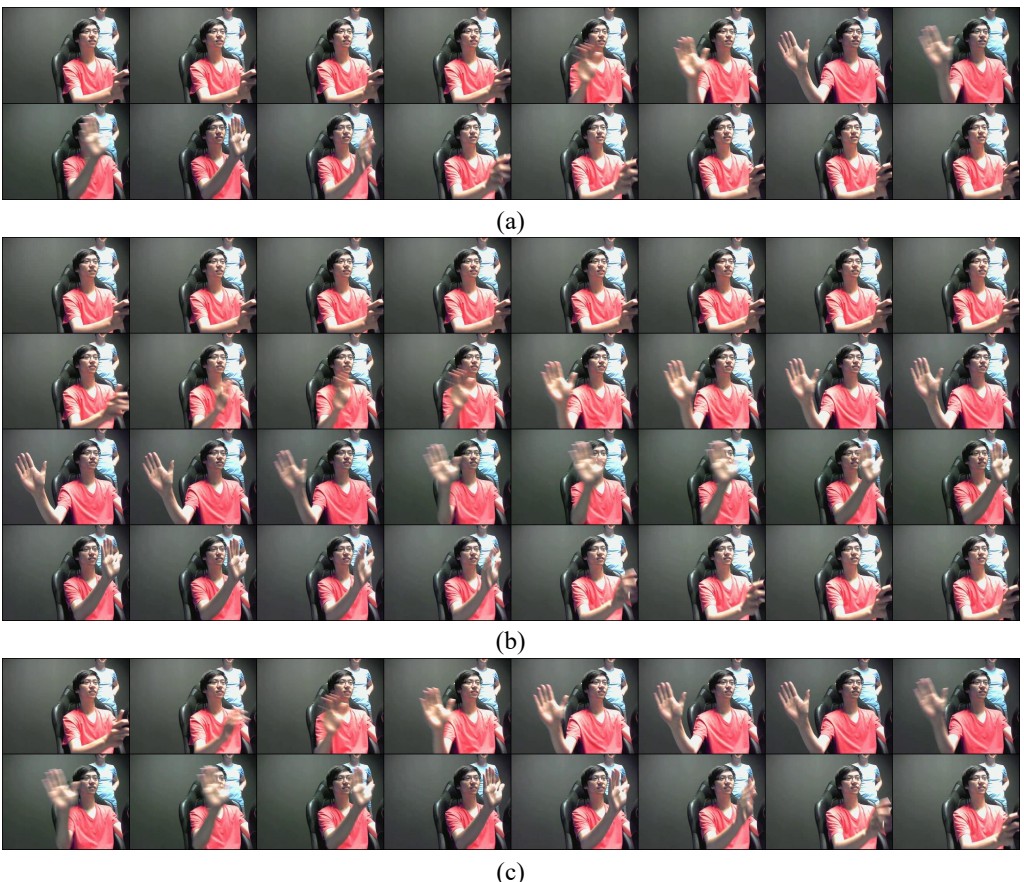

**Figure 2.** Comparison of results using different sampling strategies: (**a**) sparse temporal sampling strategy ($k = 16, n = 1$); (**b**) sparse temporal sampling strategy ($k = 32, n = 1$); (**c**) our proposed content-adaptive sampling strategy ($n = 16, step = 3$).

To sample the video sequence efficiently, we proposed a dynamic sampling strategy based on content adaptation. We use a sliding window to perform gesture detection over the incoming video frames. As shown in Figure 1, frames without gestures will not be sampled by the sampler, i.e., they cannot pass the sampler (shown as a light-colored crossed-out arrow). When the sampler detects a gesture, it starts sampling evenly (no more gesture detection) according to the time strategy until it reaches the established maximum number of frames. If the frame count of a video is less than the maximum frame count set by the sampler, we will pad the input sequence with zeros. The pseudo-code of our proposed content-based adaptive sampling algorithm can be found in Algorithm 1. Algorithm 1 can be briefly described as follows: first perform continuous gesture detection on the incoming frames and record the index of the current frames; when a gesture is detected, then stop gesture detection; the index of the gesture-containing frame and the time sampling 'step' are used to generate the final list of sampled frames. The input of Algorithm 1 is a gesture video and the output is a list of sampled frames. Figure 2c presents the sampling results of our adaptive sampler. Another role of the sampler is to act as a switch for the encoder. When a gesture is detected in the video sequence, the encoder is activated and starts receiving the input from the sampler.

Being able to detect hand gestures of various sizes is a challenge for hand detectors. In practice, we have selected the single-shot multibox detector [38] as our hand detector module because of its accuracy and real-time performance. Before feeding the sampled frames into the encoder, they need to be mapped into a sequence of tokens. In our work, we use a pre-trained DenseNet121 [10] as a 2D spatial feature extractor to map the extracted frames into meaningful features.

---

**Algorithm 1** Content-adaptive sampler.

---

**Input**: gesture video data
**Output**: frame index list

  1: initialization [frame list], counter=0, flag=True
  2: **while** cap is opened & flag **do**
  3:    **if** a gesture is not detected **then**
  4:       counter ← counter + 1
  5:       continue
  6:    **else**
  7:       flag ← False
  8:    **end if**
  9: **end while**
10: **for** each 'index' i in range(length n) **do**
11:    [frame list] ← i*step + counter
12:    i ← i + 1
13: **end for**
14: **return** [frame list]

---

### 3.2. Attention-Based Encoder

The architecture of the transformer encoder is shown in Figure 3. As shown in Figure 3a, the encoder consists of a stack of *N* identical layers. Following the suggestions in reference [21], in our work we set *N* to 6. Each layer consists of two submodules: a multi-head self-attention module, and a position-wise fully feed-forward neural network. Each submodule is connected using a residual connection (denoted as "Add" in Figure 3), followed by layer normalization. The multi-head self-attention module uses a self-attention mechanism to perform a new representation of the input sequence; the feed-forward network uses a fully connected feed-forward neural network to further transform the input vector sequence.

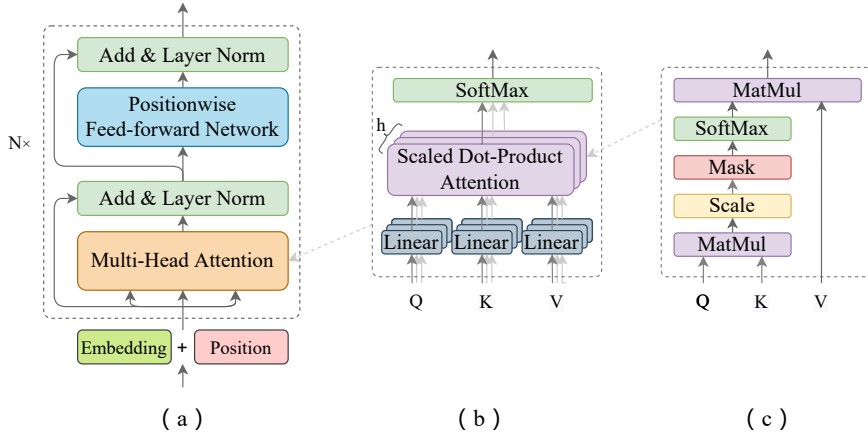

**Figure 3.** The architecture of transformer encoder: (**a**) encoder architecture; (**b**) multi-head attention; (**c**) scaled dot-product attention.

Self-Attention. Mapping a query and a group of key-value pairs to an output is the main role of the attention function. In other words, a given query can interact with the keys to guide the biased selection of values. If a key is closer to a given query, then more attention weight is assigned to the value of the key. The self-attention mechanism treats the representation of each position in the sequence as query and the representation of all positions as key and value. The self-attention model computes a weighted sum of the values of each position by calculating the degree of match between the current position and all positions, which is the attention weight in the attention mechanism. Mathematically, given

a query $q$ and $m$ key-value pairs $(k_1, v_1), \ldots, (k_m, v_m)$, the attention function $f$ is instantiated as a weighted sum of the values, which can be defined as following Equation (1) [21]:

$$f(q, (k_1, v_1), \ldots, (k_m, v_m)) = \sum_{i=1}^{m} \alpha(q, k_i) v_i \tag{1}$$

where query $q$, key $k$, value $v$, and output are all vectors. The attention weight of query $q$ and key $k_i$ is obtained by mapping the two vectors into scalars using the attention-scoring function $a$ and then by SoftMax operation:

$$\alpha(q, k_i) = \frac{\exp(a(q, k_i))}{\sum_{j=1}^{m} \exp(a(q, k_j))} \tag{2}$$

where $a$ indicates the attention-scoring function. From Equation (2), it is clear that choosing different attention-scoring function $a$ leads to different attentional behaviors. Additive attention and scaled dot-product attention are the two most commonly used attention-scoring functions. The scaled dot-product attention of queries $q$ and keys $k$ can be expressed as follows:

$$a(\mathbf{q}, \mathbf{k}) = \mathbf{q}^\top \mathbf{k} / \sqrt{d} \tag{3}$$

where $d$ denotes the length of the query $q$ and the key $k$. Since the time complexity of the self-attention operation of the original transformer model is $O(n^2)$ ($n$ is the length of the input sequence), this limits its ability to process long sequences. To address this limitation, we use an efficient self-attention variant called longformer [37] as our attention module. Longformer combines local and global attention, and its temporal complexity is linearly related to the length of the sequence, which allows long-range dependence problems to be better solved.

Multi-Head Attention. Another important technique used in the transformer is the multi-head attention mechanism. To capture the various ranges of dependencies within a sequence, we can transform queries, keys, and values using $h$ sets of linear projections learning independently. As shown in Figure 3b, these $h$ groups of transformed queries, keys, and values are then fed in parallel to the scaled dot-product attention. Finally, the outputs of $h$ scaled dot-product attention are concatenated together and transformed by another linear projection that can be learned to produce the final output. Mathematically, given a query $q$, a key $k$, and a value $v$, each attention head $h_i$ can be computed using the following Equation (4) [21] :

$$h_i = f\left(W_i^{(q)} q, W_i^{(k)} k, W_i^{(v)} v\right) \tag{4}$$

where $W$ is the weight parameter to be learned, and $f$ is scaled dot-product attention function.

Positional Embeddings. Unlike RNNs, which process video frames one by one, the self-attention is computed in parallel. Since videos are ordered sequences of frames, the transformer model needs to consider order information. To use the sequence order information, the transformer introduces positional encodings into the representation of the original input to inject relative or absolute positional information. There are various ways to calculate positional encodings, which can be obtained by learning or being fixed directly [39]. In this work, we use fixed positional encoding based on sine and cosine functions of different frequencies as shown in Equation (5) and (6):

$$PE(pos, 2i) = \sin\left(\frac{pos}{10,000^{2i/d}}\right) \tag{5}$$

$$PE(pos, 2i+1) = \cos\left(\frac{pos}{10,000^{2i/d}}\right) \tag{6}$$

where $PE()$ denotes the function of positional encodings, *pos* denotes the position, *i* represents the dimension in the positional encoding vector, and *d* is a base parameter of transformer that denotes the size of the hidden layer at each position. We do this via positional encoding. We simply embed the positions of the frames present within videos with an embedding layer. We then add these positional embeddings to the precomputed CNNs features.

MLP Head. As in BERT [27] and ViT [24], we add a special token CLS in front of each input sequence. After the input sequence is propagated through the transformer layers, the output of hidden state associated with this CLS is considered to be the final representation of the entire input sequence. Finally, this CLS token is fed to a classification MLP head for processing to obtain the final gesture prediction. The MLP blocks contain two linear layers with a GELU [40] nonlinearity and dropout [41] between them. To summarize, The input token representation is first processed by layer normalization, then encoded by the transformer, and finally the CLS tokens are sent to the MLP head to predict the results.

### 3.3. Loss Function

For the final layer in MLP used for gesture classification, the SoftMax function is normally used to convert from real-valued activations to likelihoods. The SoftMax function used in MLP can be formulated using the following equation:

$$\widehat{y}_k = \frac{\exp(a_k)}{\sum_{i=1}^{n} \exp(a_i)} \tag{7}$$

where *n* indicates the number of classes. Since the outputs are meant to be class probabilities that sum up to 1, we use the cross-entropy loss as the loss function to minimize during training. The multi-class cross-entropy loss can be expressed as the following:

$$l = -\sum_{k=1}^{n} y_k \log \frac{\exp(a_k)}{\sum_{i=1}^{n} \exp(a_i)} \tag{8}$$

where $y_k$ is the label of the correct solution, only the index of the correct solution label in $y_k$ is 1, and all others are 0.

## 4. Experiment and Results

### 4.1. Datasets

The performance of the proposed network was evaluated using two publicly available datasets, namely the NVGesture dataset [42] and the EgoGesture dataset [43,44]. Figure 4 presents some sample frames captured from these two datasets. The NVGesture dataset is a large dataset of 25 gesture types recorded by multiple sensors and viewpoints, each of which is used for human–computer interfaces. It contains a total of 1532 dynamic hand gesture videos, each containing only one gesture. We followed [42] and split the data with a ratio of 5:2:3, resulting in 750 training, 300 validation, and 482 testing videos, respectively.

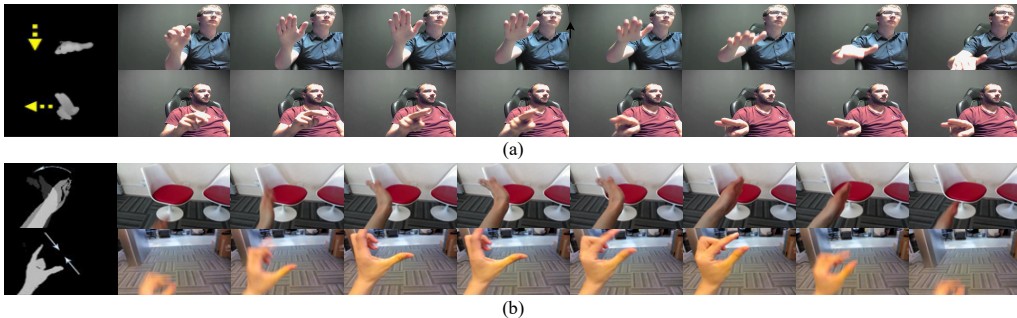

(a)

(b)

**Figure 4.** Some frames captured from two datasets: (**a**) NVGesture dataset; (**b**) EgoGesture dataset.

The EgoGesture dataset is a large-scale dataset for egocentric hand gesture recognition, consisting of 83 classes collected from four indoor and two outdoor scenes. The dataset has a total of 24,161 gesture samples recorded from 50 different subjects. The dataset was grouped in a 6:2:2 ratio, resulting in 14,416 training samples, 4768 validation samples, and 4977 testing samples, respectively. The details of the datasets used in our experiments are summarized in Table 2.

**Table 2.** The details of datasets used in our experiments.

| Data Sets | Classes | Training | Validation | Testing | Total |
|-----------|---------|----------|------------|---------|-------|
| NVGesture | 25 | 750 | 300 | 482 | 1532 |
| EgoGesture | 83 | 14,416 | 4768 | 4977 | 24,161 |

### 4.2. Evaluation Metrics

To evaluate the generalization ability of the models and compare the performance of different models, we conducted experiments using several standard evaluation metrics. Commonly used metrics for evaluating gesture recognition models are accuracy, precision, recall and F1 score. In this work, we use the standard top-1 accuracy to evaluate the prediction accuracy of proposed model. If the $\hat{y}_{i,j}$ is the predicted class for the $i$-th sample corresponding to the $j$-th largest predicted score and $y_i$ is the corresponding true value, then the top-$k$ accuracy can be defined as Equation (9):

$$\text{Top-}k \text{ Accuracy}(y, \hat{y}) = \frac{1}{n} \sum_{i=0}^{n-1} \sum_{j=1}^{k} 1(\hat{y}_{i,j} = y_i) \tag{9}$$

where $n$ is number of the samples. Precision and recall can be defined using the following equations:

$$\text{Precision} = \frac{\text{Number of True Positives}}{\text{Number of True Positives } + \text{ Number of False Positives}} \tag{10}$$

$$\text{Recall} = \frac{\text{Number of True Positives}}{\text{Number of True Positives } + \text{ Number of False Negatives}} \tag{11}$$

F1-score, also known as balanced F-score or F-measure. The best value that can be achieved for the F1 score is 1 and the worst value is 0. F1-score is the harmonic mean of precision and sensitivity, which can be expressed as follows:

$$F1 \text{ score} = 2 * \frac{\text{Precision } * \text{ Recall}}{\text{Precision } + \text{ Recall}} \tag{12}$$

where the precision and recall are calculated by Equations (10) and (11).

### 4.3. Experimental Setting

An input video is first sampled using a content-based adaptive sampler. In practice, we set the maximum number of sampled frames per video to 16 and the stride of the sampler to 3. We then resized the shorter side of all sampled frames to 256 pixels, and random crop, color jitter, and random horizontal flip were employed for data augmentation. In the inference phase, only the center crop, resizing, and normalization operations were performed on the input frames. Finally, a center crop of size $224 \times 224$ pixels was used for all frames, thus creating a batch of shape $x \in \mathbb{R}^{B \times 3 \times 16 \times 224 \times 224}$, where $B$ indicates the batch size.

The proposed network was implemented using the PyTorch deep learning framework. The model was trained and validated on a server equipped with two NVIDIA GeForce RTX 3090 GPUs. We adopted a stochastic gradient descent as the optimizer with a momentum of 0.9. We trained the model for 25 epochs with a batch size of 4. The initial learning

rate was set to 0.001 and then decayed by 0.1 times every 7 epochs. The total trainable parameters of the model are 121.28 million. Such a large model typically a requires larger training dataset to avoid overfitting. To effectively train our models on our two datasets, we employed several regularization strategies, such as data augmentation methods used in data pre-processing. In addition, we used the label smoothing mechanism [31,45] to regularize the model by estimating the marginalized effect of label dropout during training. During the testing phase, the setup is the same as in the validation phase.

### 4.4. Results and Discussion

Learning Curves. Learning curves are widely used tools for deep learning to diagnose the learning and generalization behavior of models. The accuracy and loss curves of our model during the training and validation phases are illustrated in Figure 5, where the x-axis is the time axis, the primary y-axis(left) is the value of accuracy and the secondary y-axis(right) is the loss value. As can be seen from Figure 5, after 15 epochs, the plot of validation loss decreases to a stable point and has a small gap with the training loss, which indicates that our model acquires a good fit. We observed that the gap between the validation and training loss on the NVGesture dataset was different from that on the EgoGesture dataset. In other words, our model acquires a better fit on the EgoGesture dataset than on the NVGesture dataset. The learning curve of a good fit is usually influenced by the size of the training dataset and is also related to the complexity of the model. We analyzed that the gap between the training loss and validation loss in Figure 5a is caused by the small number of samples in the training set.

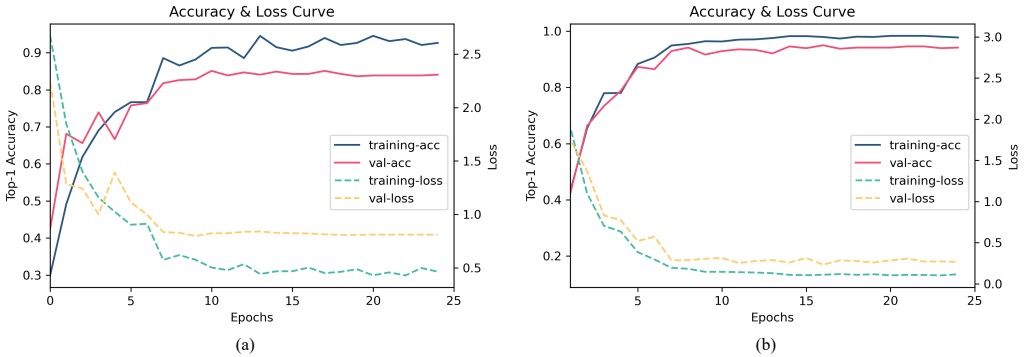

**Figure 5.** Accuracy and loss curves of the model during training and validation phases: (**a**) NVGesture dataset; (**b**) EgoGesture dataset.

Accuracy Comparison with State-of-the-Art Methods. We compared our results with the current state-of-the-art methods on NVGesture and EgoGesture datasets, and the results are summarized in Table 3. Both datasets provide corresponding color and depth videos, and in this work we evaluated the proposed model using only color videos. All these methods to be compared can be roughly divided into three categories. The first class is 2D CNN-based methods, such as TSN [14], which require pre-computation to obtain optical flow features. The second category is 3D CNN-based methods, such as R3D [42], ResNeXt [2] and C3D [16], which uses 3D convolution to learn the spatial–temporal structure of the video. The last class is the current popular 2D+1D CNN-based methods, including I3D [25] and TSM [17]. This method decouples the 3D convolution kernel to reduce the parameters of the kernel, but the spatial–temporal modeling capability of this method is limited. Unlike these three types of methods, our proposed approach is based on the attention mechanism and does not use any convolution or recurrent operations.

Table 3 shows the comparison results of recognition accuracy on two datasets. The number of the sampling frames was set to 16 for the input of all comparison methods and the same modality (color) of the test set was used. Our method achieved recognition accuracies of 83.2% and 93.8% on NVGesture and EgoGesture datasets, respectively. From Table 3, it can be seen that the proposed approach achieves the best recognition performance

compared to other methods. To measure the influence of our proposed adaptive sampler on the performance of the model, we conducted ablation experiments on two datasets. As shown in the last two rows of Table 3, our model achieves 80.7% and 92.6% accuracy on NVGesture and EgoGesture datasets, respectively, when using only the attention mechanism. After improving the sampling using the proposed sampler, the accuracy of the model was improved by 2.5% and 1.2%, respectively. This illustrates the effectiveness of our proposed sampler in improving the accuracy of model recognition.

**Table 3.** Accuracy comparison with other methods on the test sets of NVGesture and EgoGesture datasets.

| Methods | Models | NVGesture | EgoGesture |
|---------|--------|-----------|------------|
| TSN [14] | Two-stream | 65.6% | N/A |
| R3D [42] | 3D CNNs | 74.1% | 78.4% |
| C3D [16] | 3D CNNs + SVM | 69.3% | 86.4% |
| ResNeXt [2] | 3D CNNs | 66.4% | 90.9% |
| I3D [25] | Inflated 3D CNNs | 78.3% | 90.3% |
| TSM [17] | 2D + 1D CNNs | N/A | 92.1% |
| Ours | Attention | 80.7% | 92.6% |
| Ours | Attention + Sampler | 83.2% | 93.8% |

F1-score. The F1 score is a common metric for measuring the precision of gesture recognition models. The F1-score can be defined as the harmonic mean of the precision and recall. We counted the recall, precision and F1 score of our proposed model on both NVGesture and EgoGesture datasets, and the results are summarized in Table 4. As can be seen from Table 4, our proposed attention-based model achieves F1 score of 82.98% and 93.10% on the NVGesture and EgoGesture datasets, respectively.

**Table 4.** Precision, recall, and F1-score of our proposed model on two datasets.

| Data Sets | Precision | Recall | F1-Score |
|-----------|-----------|--------|----------|
| NVGesture | 83.56% | 83.13% | 82.98% |
| EgoGesture | 93.92% | 93.82% | 93.10% |

Model Complexity. When evaluating a deep model for gesture recognition, the complexity of the model should be considered in addition to performance metrics. The number of parameters and floating-point operations (FLOPs) are two commonly used indicators to evaluate the complexity of a deep model. The number of parameters describes the storage space required to store the model, and FLOPs describe the computational force required to use the model. We evaluate the complexity of our model by measuring the number of parameters and FLOPs, and the results are shown in Table 5. We report the differences in model parameters, FLOPs and processing time when 8, 16, 24 and 32 frames are used as sampling lengths. As can be seen from Table 5, as the sampling length increases, the computation and processing time of the model increases accordingly.

**Table 5.** Parameters, floating-point operations and inference time of our proposed model.

| Input Shape | Parameters (M) | FLOPs (G) | Processing Time (ms) |
|-------------|----------------|-----------|----------------------|
| (1, 3, 8, 224, 224) | 121.277 M | 189.877 G | 23.5 ms |
| (1, 3, 16, 224, 224) | 121.284 M | 379.698 G | 46.3 ms |
| (1, 3, 24, 224, 224) | 121.290 M | 569.519 G | 67.2 ms |
| (1, 3, 32, 224, 224) | 121.296 M | 759.340 G | 73.4 ms |

Confusion Matrix. To evaluate the recognition accuracy of the proposed network for each type of gesture, we computed the confusion matrix table. The confusion matrix was obtained by comparing the predicted labels and the ground truth labels. We visualized the confusion matrix table obtained on two datasets, as shown in Figure 6. The values on the diagonal line represent the number of gestures correctly recognized by the model, while the values on the off-diagonal line represent the number incorrectly recognized. The value can also be distinguished based on the shade of the color. The higher diagonal value indicates the better performance of the model. In Figure 6, we use different colors to indicate the magnitude of the values separately. The confusion matrix shows that the proposed network is robust to each gesture class.

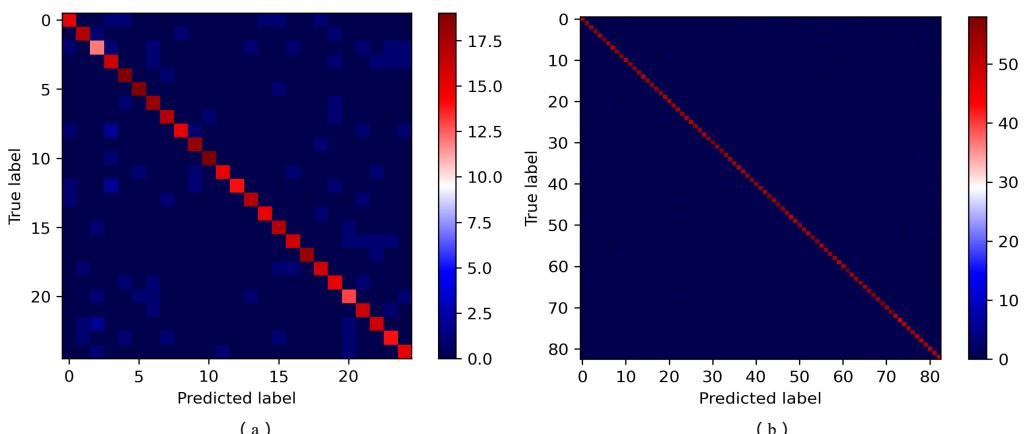

**Figure 6.** The confusion matrix of the gesture recognition results using our method on two datasets: (**a**) NVGesture dataset (25 gesture actions); (**b**) EgoGesture dataset (83 gesture actions).

## 5. Conclusions

In this work, we present a network based on content-adaptive and attention mechanisms for hand gesture recognition. The proposed network is a pure self-attention mechanism-based architecture, which does not have any convolution or recurrent operations. The self-attention mechanism has the advantage of both shortest maximum path length and minimum sequential operation sequential operation. The properties of the self-attention mechanism give the network the ability to classify gestures by focusing on information from the entire video sequence. To improve the sampling of the model and reduce the input of gesture-free frames, we introduced a content-based adaptive sampler. We use a sliding window for gesture detection on the incoming frames to determine the starting sampling frame for the encoder. We evaluated the proposed network on two publicly available benchmark datasets, NVGesture and EgoGesture. The extensive comparative experimental results demonstrate that the proposed hand gesture detection and recognition method outperforms the existing approaches based on CNNs or RNNs. In the future, we plan to further improve the recognition accuracy of the attention-based model on gesture recognition tasks. The study of uncertainty in the attention-based model will also be our future work.

**Author Contributions:** Conceptualization, Z.C.; methodology, Z.C. and Y.L.; software, Z.C.; validation, Z.C. and Y.L.; investigation, Z.C. and Y.L.; writing original draft preparation, Z.C.; writing review and editing, Z.C., Y.L. and B.-S.S.; supervision, Y.L. and B.-S.S.; project administration, B.-S.S.; funding acquisition, B.-S.S. All authors have read and agreed to the published version of the manuscript.

**Funding:** This work was supported by the Institute of Information and Communications Technology Planning and Evaluation (IITP) grant funded by the Korean government (MSIT) (No. 2020-0-00594, Morphable Haptic Controller for Manipulating VR·AR Contents), and in part by the China Scholarship Council.

**Institutional Review Board Statement:** Not applicable.

**Informed Consent Statement:** Not applicable.

**Data Availability Statement:** Not applicable.

**Conflicts of Interest:** The authors declare no conflict of interest.

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
