# Peer review of "Content-Adaptive and Attention-Based Network for Hand Gesture Recognition"

_applsci, doi:10.3390/app12042041_

Round 1

Reviewer 1 Report

Decision: Major Revision

The idea presented in this paper is worthy and can be proved beneficial in terms of an effective approach for gesture recognition by transformer-based structure for the hand gesture recognition task, which is a completely self-attentional architecture. The presentation of the paper is not convincing there are some major and minor comments and suggestions for improvement of this paper which should be addressed for possible acceptance in this journal. I looking forward to see these changes in the revised version.

Major/Minor points

  1. Abstract: In the said paper authors should add and mention the name of datasets to avoid confusion for readers. Similarly, authors should add numerical improvements in the
  2. More suitable keywords should be selected and should be sorted alphabetically.
  3. Introduction Section: The current challenges are not crystal clearly mentioned in the introduction section of this paper. I suggest adding a dedicated paragraph about the current challenges of this area followed by the authors’ contribution to overcoming those challenges.
  4. The manuscript, however, does not link well with recent literature on recognition that appeared in relevant top-tier journals, e.g., the IEEE Intelligent Systems department on " att-net: Enhanced emotion recognition system using lightweight self-attention module". Also, new trends of AI for recognition “mlt-dnet: recognition using 1D dilated CNN based on multi-learning trick approach” are missing it should be comprised.
  5. Experiment Section: The paper lacks enough details about different methods, comparisons between different methods, datasets, and other related info - details can be provided, and how fair comparisons can be achieved, should be explained.
  6. The performance of the proposed method should be better analyzed, commented and visualized in the experimental section (need a comparison table with recently published article).
  7. Please discuss the hyperparameters setting of the proposed model and other comparable models.
  8. The complexity of the proposed model and the model parameter uncertainty is not mentioned.
  9. The “Discussion” section should be added in a more highlighting, argumentative way. The author should analyze the reason why the tested results are achieved.
  10. The readability and presentation of the study should be further improved. The paper suffers from language problems.
  11. Section Conclusion - Authors are suggested to include in the conclusion section the real actual results for the best performance of their proposed methods in comparison towards other methods to highlight and justify the advantages of their proposed methods.

Reviewer 2 Report

This article aims to solve the problem of hand gesture recognition by using a transformer and attention mechanism. The authors propose a transformer-based deep learning network architecture and describe the key components of the model and special methods for processing hand gesture image data. The experimental results also show good accuracy results. All in all, this is an in-depth study. However, there are still some deficiencies in this paper, and it is recommended that the author revise the following parts to improve the quality of the article.

  1. The paper lacks experimental results to verify that the adaptive sampling strategy claimed in the abstract and conclusion can effectively reduce computational consumption. There is also a lack of relevant experimental methods and experimental evaluation metrics (such as GPU usage, memory usage, storage space, computing time, etc.) and a lack of comparison with other recent SOTA methods.
  2. It is recommended that the authors provide a comparison table of existing literature to clearly demonstrate the differences and innovations between the proposed content-adaptive and attention-based network and related works.

  3. What is the meaning of the different forms of arrows in Figure 1?  (e.g. Arrows are crossed out?) What is the difference between the arrows with dark colors and light colors? The author should have a corresponding description in the main texts.

  4. The pseudocode of Algorithm 1 also lacks relevant descriptions, the important steps in the pseudocode should be explained.

  5. The paper also lacks trained content-adaptive and attention-based network model architecture visualization plots. Please add it on.

  6. Equation 1 and Equation 3 should add citations.

  7. The prediction accuracy of the model proposed in the paper is only 93% at the highest, and why is there still a 7% error? How to solve it in the future? The authors may provide a further in-depth discussion about this part.

  8. Other models compared in Table 2 lack references. Citations should be added.

  9. What is the deep learning framework for the model development proposed in the paper? Is it PyTorch, Tensorflow, MXNet, or some other framework? Please provide additional explanations in the main text.

  10. Table 3 lacks comparison results with other SOTA methods. The paper should not be limited to the experimental analysis and compare results of the proposed model on the two data sets, which cannot yet demonstrate the superiority over other SOTA methods.

Reviewer 3 Report

The manuscript entitled “Content-Adaptive and Attention-Based Network for Hand Gesture Recognition”, proposed a real-time hand gesture recognition system based on the transformer encoder network. The author introduced an adaptive sampling strategy to reduce the input of gesture-free frames. However, I believe, a revision is necessary before accepting the manuscript.

  • The proposed method is not clear. Please draw a network diagram for better understanding. The author should describe the network parameters.
  • The accuracy and loss curve is not satisfactory. There is a huge gap between them. Is there any other way to improve the accuracy?
  • The author used two publicly available datasets. Please compare your model with others.
  • Figure 6(a) and Figure 6(b) looks different. Please uniform both figures.
  • There is too much introductory writing in the abstract section. Make it concise and meaningful.
  • There are too many repeated words and sentences. For example, this sentence is repeated in abstract, introduction, and related work – “The input of the recurrent neural network currently depends on the output of the hidden state at the previous moment, thereby limiting its parallel computation.”. And many more.
  • The author mentioned in the abstract – “The input of the recurrent neural network currently depends on the output of the hidden state at the previous moment, thereby limiting its parallel computation.”. The word “currently” is confusing and technically wrong. Please rewrite the sentence.
  • The introduction and the related works are almost similar. Rewrite the introduction and related work part.
  • The motivation of this work is missing. The author may introduce some applications such as gesture-based writing systems [1], air-writing [2], sign language [3], etc.
    1. https://ieeexplore.ieee.org/document/9404361
    2. http://ieeexplore.ieee.org/document/7322243/
    3. https://www.mdpi.com/1424-8220/21/17/5856
  • The “Data preprocessing” subsection in 4.3 need to move to the proposed method section (3).
  • In section 4.4, “the plot of validation loss decreases to a stable point and has a small gab with the training loss”. What does it mean by “gab” here?

Round 2

Reviewer 1 Report

The authors successfully addressed my comments and suggestions. Good Luck!

Reviewer 2 Report

I am satisfied with the content of the author's revision, and the paper has met the requirements for publication.

Reviewer 3 Report

The manuscript entitled “Content-Adaptive and Attention-Based Network for Hand Gesture Recognition”, proposed a real-time hand gesture recognition system based on the transformer encoder network. The author introduced an adaptive sampling strategy to reduce the input of gesture-free frames. 

The author carefully revised the manuscript and I believe it is far better than before. I recommend accepting the manuscript.